# Exploring contextual adaptations in caregiver interventions for families raising children with developmental disabilities

Zsofia Szlamka[1]*, Charlotte Hanlon[2,3,4], Bethlehem Tekola[1], Laura Pacione[5,6], Erica Salomone[5,7], WHO CST Team[5¶], Chiara Servili[5], Rosa A. Hoekstra[1]

1 Department of Psychology, Institute of Psychiatry, Psychology & Neuroscience, King's College London, London, United Kingdom, 2 Centre for Global Mental Health, Department of Health Services and Population Research, Institute of Psychiatry, Psychology & Neuroscience, King's College London, London, United Kingdom, 3 Department of Psychiatry, WHO Collaborating Centre for Mental Health Research and Capacity-Building, School of Medicine, College of Health Sciences, Addis Ababa University, Addis Ababa, Ethiopia, 4 Centre for Innovative Drug Development and Therapeutic Trials for Africa, Addis Ababa University, Addis Ababa, Ethiopia, 5 Department of Mental Health and Substance Use, World Health Organization, Geneva, Switzerland, 6 Department of Psychiatry, Division of Child and Youth Mental Health, University of Toronto, Toronto, Ontario, Canada, 7 Department of Psychology, University of Milano-Bicocca, Milan, Italy

¶ Membership of the WHO CST Team is provided in the Acknowledgments.
* zsofia.szlamka@kcl.ac.uk

**Data Availability Statement:** All relevant data are within the manuscript and its Supporting information files.

## Abstract

There are increasing efforts to scale up services globally for families raising children with developmental disabilities (DDs). Existing interventions, often developed in high income, Western settings, need substantial adaptation before they can be implemented in different contexts. The aim of this study was to explore perspectives on the role that context plays in the adaptation and implementation of interventions targeting caregivers of children with DDs across settings. The study question was applied to the Caregiver Skills Training (CST) programme of the World Health Organization specifically, as well as to stakeholder experiences with caregiver interventions more broadly. Two focus group discussions (FGDs; n = 15 participants) and 25 individual semi-structured interviews were conducted. Participants were caregivers of children with DDs and professionals involved in adapting or implementing the CST across five continents and different income settings. Data were analysed thematically. Four main themes were developed: 1) Setting the scene for adaptations; 2) Integrating an intervention into local public services; 3) Understanding the reality of caregivers; 4) Challenges of sustaining an intervention. Informants thought that contextual adaptations were key for the intervention to fit in locally, even more so than cultural factors. The socio-economic context of caregivers, including poverty, was highlighted as heavily affecting service access and engagement with the intervention. Competing health priorities other than DDs, financial constraints, and management of long-term collaborations were identified as barriers. This study validates the notion that attention to contextual factors is an essential part of the adaptation of caregiver interventions for children with DDs, by providing perspectives from different geographical regions. We recommend a stronger policy and research focus on contextual adaptations of interventions and addressing unmet socio-economic needs of caregivers.

**Funding:** This study was part of Zsofia Szlamka's PhD project. It was funded by the Medical Research Council's Doctoral Training Programme (Funder reference number: 1658511) in the United Kingdom. Her attendance in the WHO CST Consultation Meeting in Xiamen, China was funded by Autism Speaks. The funders had no role in study design, data collection and analysis, decision to publish, or preparation of the manuscript.

**Competing interests:** The authors have declared that no competing interests exist.

## Introduction

Developmental disabilities including autism and intellectual disability are lifelong conditions involving difficulties in the acquisition and execution of intellectual or social functions [1]. While there is considerable variation in prevalence estimates of DDs due to methodological variation, the worldwide prevalence of intellectual disability is at least 1% [2], but may be as high as 1.5% [3] to 1.9% [4]. Similarly, the global prevalence of autism is estimated to be around 1% [5]. The global burden of DDs disproportionately affects low-and middle-income countries (LMICs): a recent global analysis suggests that 95% of children with DDs live in LMICs [4]. In these countries access to support programmes is often limited or non-existent [6, 7]. Moreover, most of the available interventions come from high-income settings and evidence of programme effectiveness in LMICs is sparse [8]. Barriers to receiving support exist at many levels: from accessibility, availability and affordability of a programme, to the acceptability for beneficiaries [9, 10].

A promising approach to developing sustainable support for families is by training caregivers [6]. Caregivers can be mediators of interventions: this approach promotes that children with DDs learn skills in different environments, at the venue of the interventions as well as at home [11–13]. It also increases caregivers' self-confidence in supporting their children [11–13]. The focus of caregiver interventions ranges from psychoeducation (to increase caregiver knowledge) to skills training (teaching caregivers strategies to support their child's skills development) [14]. There are increasing efforts to develop and then scale up such caregiver interventions in LMICs [15–17]. As part of these efforts, a frequently used method addressing shortages of specialist health resources is task-sharing: a strategy of training non-specialists to deliver circumscribed interventions while supported by more specialist workers [18]. Task sharing is suggested to help increase access to care from high- to low-income settings [19]. However, critiques proposed that task sharing may not always be acceptable and feasible in lower resource contexts: non-specialists may not feel competent to deliver certain tasks for example [20]. There are several examples of interventions for DDs developed or adapted for use in LMICs [21, 22], including a caregiver intervention using family networks in Pakistan [23]; a lay health worker delivered programme for autism in India [24], and two parent-mediated interventions in India and Pakistan [24–27].

Building on the efforts described above, an example of developing and scaling up a caregiver intervention globally is the Word Health Organization's (WHO) Caregiver Skills Training (CST). This programme is evidence-informed and has grown out of the collaboration of WHO with Autism Speaks and other international partners [28]. Following WHO's aim to scale up freely available, evidence-based interventions for people with mental health conditions, the CST was designed to be adapted and feasible in low-resource settings [29]. The CST is a low-intensity training intervention for caregivers of children between two and nine years of age with DDs or with developmental delays who may not (yet) have received a formal diagnosis. The CST consists of nine sessions delivered to groups of caregivers and three home visits and is manualised by facilitator guides and participant booklets [29]. The programme is intended to be integrated into existing public maternal, child, and family health and social services. To inform adaptations to the WHO CST programme tested in countries across the world, an adaptation guide was developed by WHO. This guidance also delineates a recommended adaptation process based on the work of Wild [30]. The suggested process includes creation of a local adaptation team, formal consultation with an adaptation advisory group of community stakeholders, including caregivers and local cultural and religious leaders where appropriate. Adapted versions of the CST are currently undergoing pilot implementation in several countries [28]. Details of the WHO CST's adaptation process can be found in S1 File.

Contextual factors may shape the way an intervention works in different settings, how it is implemented and impact on the outcomes achieved [31, 32]. Therefore local adaptation is recommended to ensure that an intervention remains effective across settings, while meeting the needs and expectations of local communities [33]. In the existing literature there is no clear agreement as to what *context* means. Some studies consider contextual changes to be part of the implementation and not the adaptation process [34, 35]. Many scholars agree that contextual adaptation goes beyond the geographic setting [36, 37]. Previous studies mentioned examples such as activity types in the intervention and broader societal discourses about the adaptation and implementation process [32, 38]; organisational and community norms and attitudes [39]; or the physical and environmental surroundings of the intervention, socio-economic factors and fidelity [40]. Meanwhile in the adaptation literature there has been a strong focus on cultural adaptations as compared to other contextual factors [30, 31, 41–46]. An example of this cultural focus is using the term 'ubuntu' to ground early interventions for DDs in Africa [47]. In this paper we define context as outlined in UK Medical Research Council's complex intervention framework for health systems research: including broader discourses, infrastructural relations, institutional relations, history of relations across stakeholders, individual interactions across stakeholders, and individual acts within intervention delivery [36]. We understand culture as one of the elements of broader discourses contributing to context.

Interventions that are contextually adapted to meet the needs of participants have been shown to be effective and have improved feasibility [17, 41–43]. However, it remains unclear what contextual adaptations should involve and what role culture plays in the adaptation process. There can be a tension between the need to retain evidence-based elements of interventions while making sure that an intervention is contextually appropriate [48]. It is also unclear at what point an adaptation becomes so substantial that it results in a de novo intervention [49].

## Aim of the study

The aim of this study was to investigate the role of context in adapting and implementing caregiver interventions for families raising a child with a DD across the globe.

## Methods

We conducted a qualitative study using a phenomenological approach [50]. This approach was chosen to allow for the investigation of individual experiences with caregiver interventions as well as to explore how participants perceive the role of context in caregiver interventions. There were two phases to the study. Phase 1 focused specifically on experiences with the WHO CST programme, while Phase 2 considered experiences with different caregiver interventions more broadly.

### Phase 1

Two focus group discussions (FGDs) were conducted during a technical consultation meeting of the WHO CST in Xiamen, China on 8-9th November 2018, an event attended by 42 participants from twenty country teams adapting and implementing CST and leads of CST in WHO and Autism Speaks.

**Participants.** All participants attending the WHO CST meeting were invited to take part in FGDs. Overall, fifteen participants consented to participate, ten in the first and five in the second group, representing six country teams of the twenty countries present. Participants were clinicians, researchers and caregivers involved as team leads or team members in the adaptation of CST (details in Table 1). In both phases participants were assigned to stakeholder

**Table 1. Characteristics of participants in the focus group discussions.**

| Region | Overall | Clinicians | Caregivers | Researchers |
|---|---|---|---|---|
| Africa | 1 | n/a | n/a | 1 |
| Americas: | | | | |
| North America | 1 | n/a | n/a | 1 |
| Central and South America | 8 | 5 | 3 | n/a |
| South-East Asia | 1 | 1 | n/a | n/a |
| Europe | 0 | n/a | n/a | n/a |
| Eastern Mediterranean Region | 0 | n/a | n/a | n/a |
| Western Pacific | 4 | 2 | 1 | 1 |

group categories according to what they identified their main role to be. Some participants may have hold dual roles (e.g. a clinician with some involvement in research). While all participants were familiar with the adaptation of CST, they differed in terms of their practical experience. In FGD1 most stakeholders had already completed the CST adaptation process while participants in FGD 2 were only just starting it. Clinicians worked in or ran non-governmental organisations (NGOs), others worked in public hospitals. Researchers were affiliated to universities. Caregivers had overlapping experiences with working on adaptations, participating in caregiver interventions, and even running their own caregiver association. Attendance during the WHO CST meeting was not geographically representative of all continents and this is reflected in participation in the FGDs.

## Phase 2

In the second phase individual interviews were conducted with members of CST teams, caregiver organisations and NGOs working closely together with CST teams globally. The aim of the interviews was to follow up on themes developed in Phase 1 and triangulate across methodologies and types of respondent [51]. The interviews also touched on topics applicable to a broader range of caregiver interventions than the CST programme alone.

**Participants.** Participant recruitment for individual interviews took place by inviting members of the CST country teams. Participants were also recruited through snowballing: CST team members connected the research team to other potential participants who had experience with adapting or participating in caregiver interventions (CST or other programmes). Overall, twenty-five interviews took place using online audio or videocalls (details in Table 2): twenty-three in English and two in Spanish; representing ten countries. Eleven participants from six CST country teams had previously participated in the FGDs. Participants were

**Table 2. Characteristics of participants in the semi-structured interviews.**

| Region | Overall | Clinicians | Caregivers | Researchers | International organisations |
|---|---|---|---|---|---|
| Africa | 2 | 1 | n/a | 1 | n/a |
| Americas: | | | | | |
| North America | 4 | n/a | n/a | n/a | 4 |
| Central and South America | 11 | 6 | 3 | 1 | 1 |
| South-East Asia | 1 | 1 | n/a | n/a | n/a |
| Europe | 1 | 1 | n/a | n/a | n/a |
| Eastern Mediterranean Region | 1 | | n/a | n/a | 1 |
| Western Pacific | 5 | 3 | n/a | 2 | n/a |

clinicians, researchers, caregivers of children with DDs or representatives of WHO or Autism Speaks. Participants represented NGOs; universities; international organisations; and advocacy organisations. Interviewing WHO and Autism Speaks representatives helped data source triangulation [52] and provided a global perspective on the research questions.

**Interview guides.** The FGD topic guide was developed based on previous literature and was piloted before implementation. Stakeholder views were explored on experiences with running or participating in caregiver interventions, both in general, and with a specific focus on the WHO CST. The interview guide for CST country teams, caregivers and NGO representatives was based on preliminary findings from the FGDs and was adjusted and further developed iteratively after each interview. A separate topic guide was developed for participants from WHO and Autism Speaks. Representatives of these organisations led the development of the programme materials and provided technical advice and assistance to country teams but were not working directly on implementation locally. Along with interviews, contextual information and reflections about interviews were noted in an interview diary. All interview guides can be found in S2–S4 Files.

**Researchers' positionality.** FGDs were conducted by ZS with support from RAH. ZS is a PhD student in Psychology, RAH is an academic Psychologist researching autism in low-resource settings, and leads research on the adaptation and evaluation of the CST programme in Ethiopia [80]. Neither ZS nor RAH is clinically qualified. The FGDs took place during a meeting facilitated by WHO which might have framed the conversations: the way ZS and RAH were perceived by informants and the extent to which informants felt comfortable discussing sensitive topics. For example, the formality of the event might have prevented some participants from sharing experiences comfortably. On the other hand, both ZS and RAH had previously met some of the informants: this created a stronger rapport with some participants.

During interviews, some participants thought that ZS was part of the WHO CST team that developed the intervention. This may have influenced what information they felt comfortable sharing. In such cases ZS clarified that she was not part of the WHO CST team. Other participants expressed that the fact that ZS was not part of the core team provided a possibility to discuss critiques about CST.

**Patient and participant involvement.** A document summarising the preliminary results from the FGDs were sent to participants for feedback and reflection. Participant feedback then informed the development of individual interviews.

## Analysis

Transcription was done by ZS, taking a naturalised approach in order to represent as much as possible from the research environment and reflect a verbatim depiction of speech [53, 54]. Data were anonymised and analysed using reflexive thematic analysis with the qualitative data management software NVivo 12 [55]. Thematic analysis was chosen because of its flexibility and the exploratory nature of the study [56].

Coding started upon completion of the FGDs. The first coding of FGDs was done by ZS and in discussion between ZS, RAH and CH, allowing for investigator triangulation [52]. Upon completion of the individual interviews, data from Phase 1 and Phase 2 were merged and the inductive analysis continued using the whole dataset. The first three individual interviews and the two FGDs were also coded by a second coder (BT), who is an Ethiopian Social Scientist with has extensive experience in qualitative research and who has worked previously on the adaptation of CST in Ethiopia [47]. The two coders then shared their codebooks, discussed preliminary themes they developed and agreed on a codebook. ZS coded the remainder of the interviews and further refined the themes and subthemes based on discussion with RAH

and CH. The iterative theme development was also informed by literature on what contextual adaptations mean as compared to cultural adaptations [36]. After the main themes were developed, they were compared across regions and stakeholder groups. During this process, ZS undertook deviant case analysis to enhance the rigour of the study [57]. As part of this, ZS reread the transcripts after having developed the main themes to see whether any stakeholder perspectives could not be accounted for within the analysis.

## Ethical considerations

The study received ethical approval from the Psychiatry, Nursing & Midwifery subcommittee of King's College London's College Research Ethics Committee (reference LRS-18/19-8447 for FGDs and RESCM-18/19-8447 for interviews). Our ethical approval allowed for data collection for Phase 1 of this study with participants on the premises of the WHO CST Consultation Meeting that took place in China. All participants provided written informed consent.

## Results

Four main themes were developed: 1) Setting the scene for adaptations; 2) Integrating an intervention to local public services; 3) Understanding the reality of caregivers; 4) Challenges of sustaining an intervention. Each theme will be discussed below, with representative quotes illustrating the findings. All relevant quotes for each theme can be found in S1 Table and a summary of themes in Table 3 below.

### Setting the scene for adaptations

Informants all expressed the view that any caregiver intervention should be adapted contextually to the setting before implementation. Many participants agreed that local adaptations can be done on different levels: adapting psychological concepts used in the intervention and adapting elements such as illustrations and the way in which translation is done.

**The role of culture in adaptations.** Speaking about caregiver interventions in general, participants had different perspectives on the impact culture has on the effectiveness of an intervention and the depth of adaptations needed. Some thought that the strategies often included in caregiver interventions to support caregivers can be applied universally, while others argued that intervention materials and strategies require more extensive adaptations to

**Table 3. Summary of main themes and sub-themes.**

| Main theme | Sub-themes |
| --- | --- |
| Setting the scene for adaptations | The role of culture in adaptations |
| | The expectations and attitudes of caregivers |
| | Technical aspects of adaptations: Psychological concepts; translations; culturally representative illustrations; using online tools |
| Integrating an intervention to local public services | Contacting caregivers who may need support |
| | Adapting to local practicalities |
| | Settings with competing services |
| Understanding the reality of caregivers: | The location and venue of the programme |
| | Education and literacy of caregivers |
| | Supporting families in poverty |
| Challenges of sustaining an intervention | Supporting families in the context of competing priorities |
| | The role of funding in sustaining a programme |
| | Coordinating long-term stakeholder collaborations |

meet local needs. Many expressed the view that contextual factors other than culture have a more immediate impact on adaptations than culture itself. Informants from international organisations highlighted that adaptations should be both contextual and cultural. A few added that the extent to which cultural adaptation was needed depends on how far removed the setting is from the culture in which the programme was initially developed.

*PCP202, clinician, Americas*

*"For sure the answer is yes [culture is important], but I think also an important thing to consider is how different are the cultures in which the intervention was developed and the intervention that you want to implement. . . I would think that in our context we are pretty much occidental so in general interventions. . . developed in Europe may be easy to adapt."*

**The expectations and attitudes of caregivers.**   Participants talked extensively about the role that expectations and attitudes play when adapting and implementing a programme. They argued that stigma can affect attitudes towards DDs and caregiver engagement with an intervention. They put forward that stigma can come both from the wider community or stem from the caregivers' own attitudes and feeling of shame and guilt. Some participants mentioned that in their settings spiritual or religious explanations are common regarding why DDs develop, and these issues need to be addressed for an intervention to be effective.

*PCP6, clinician, Western Pacific*

*"I think one adaptation that I will talk about in the next meeting. . . is about the stigma. . . the myth related to the origin of the disease. Because some of [. . .] the [CST training] manual is quite like ancient, or spell or something and I don't think people here talk about that anymore, so I will focus more on the shame and guilt in mothers. . .not about you know the spell, the magic."*

Informants across settings thought that caregivers may have unrealistic expectations about what the intervention can achieve and what is feasible given the resources. Examples they gave from their experiences with CST include the expectation that the intervention would cure the child with an DD; that caregivers would like to receive immediate help in the form of a quick intervention that brings about visible change. Participants from the Western Pacific added that this can be reinforced in a context in which caregivers have various work commitments and are living under constant time pressure. Others mentioned that in their setting caregivers expect expert knowledge in managing their child's symptoms and developing their skills. They may therefore find it hard to accept a caregiver intervention where they are expected to be the ones scaffolding their child's early development. Informants recommended to clarify caregiver expectations and how professionals can respond to them to make the adaptation process more successful.

*PCP20, clinician, Western Pacific*

*"We know that the parents always think that the professional will do better than me, so they tend to bring their children to clinics and let the therapist teach their child. So when they come to the CST, even though that we have already taught them in this model that you have to learn by yourself but still there are some parents very surprised: oh I need to play, I need to work with my child by myself. . . and then one or two mothers drop out because she thinks that this is not what she wants."*

**Technical aspects of adaptations.**    Informants discussed programme elements and aspects that they believe are important to adapt, based on their experience with adaptation and implementation of CST and other caregiver interventions.

*Psychological concepts.* Informants raised that some psychological concepts core to the intervention can be difficult to translate across settings and many gave the example of the term 'joint engagement'. Many thought that the concept of joint engagement was difficult to understand for caregivers. Some believed that it was because of cultural differences in play. They raised that engagement through play in which they caregivers and children act as equal partners does not come naturally to caregivers in their context. Such differences can mean that some intervention strategies are not easy to understand for caregivers and therefore harder to implement.
*PCP20, clinician, Western Pacific*

*"They [caregivers] went through lots of practice. . .they finally find out oh, so this is engagement. . .I think it's because in our culture our parents do not think that adults and a child is in equal status. Because they always think that. . .because I'm the adult, I ask you to do something, you have to do [it]. Like I ask you to wash your hands and then you wash your hands. I ask you to eat this, you eat this. So most of our parents think that they can control the interaction with their children."*

*Translations.* Most informants agreed that translation to the local language or dialect is essential so that caregivers could easily understand the programme. Many participants thought that a term may exist in one language but not in another. These concepts may then need to be explained in more detail or in the form of a full sentence in the adapted materials. Informants emphasised the need to capture the spirit of the text rather than literal translation.
*PCP123, clinician, South-East Asia*

*"For instance, we say 'catch the child being good' right? When you translate it in our local language it's like 'catch him!'. . .maybe you back translate it, it translates well, but it loses the essence of what we are trying to say. It doesn't mean to just catch him physically, right?"*

*Culturally representative illustrations.* In general, participants thought that visuals such as the illustrations included in the CST training materials could help caregivers with less education understand the intervention better. They had different views on whether to make the visual parts of materials culturally representative. Some found it important to have illustrations that the local community could identify with. Others expressed that it might help caregiver engagement and spread to other sites if pictures remained globally applicable.
*PCP123, clinician, South-East Asia*

*"We have people who are very dark-complexioned and very fair and those who have more South-Eastern features. We can relate to a lot of those pictures, but not everyone relates to every picture."*

*Using online tools.* Some participants mentioned that online delivery techniques for caregiver interventions could help expand access to people from a wider range of locations, address costly transportation, geographically inaccessible locations, and working in remote areas. Some participants highlighted that caregivers who could not travel long distances could attend the intervention online, therefore making service access more equitable. However, participants added that local constraints should be considered before opting for online solutions, as the example below from the CST shows.

*PCP205, caregiver, Americas*

*"We are starting to have a web app [for CST and other materials for caregivers], because people in our setting don't have those smartphones with a lot of memory, so they won't engage. . .they won't spend memory on an app, so we are starting to do a web app and put videos, like YouTubers, from master trainers with the tips from each session."*

## Integrating an intervention into local public services

Many participants discussed the importance of integrating the adapted intervention within existing public health, educational or social services.

**Contacting caregivers who may need support.**    Informants talked about ways in which caregivers in need can be identified and informed about possible interventions. Participants working in schools raised that caregivers may get worried and feel stigmatised if they receive a targeted invitation to participate.

*PCP101, clinician, Americas*

*"If we are doing it [caregiver intervention] in community centres that are in schools. . .if we tell the teachers to choose kids that have some challenge in development, then parents would feel that the school is maybe trying to expel them or that they are doing something wrong. And they have no diagnosis, so some of them are not even worried. . .and then somebody is telling them you have to worry, you are not seeing something. . .if you give a speech on development [awareness raising programme] for everybody and say that whoever has a worry on any of these aspects can come to this very special, exclusive intervention for free and then people would queue up to do it."*

**Adapting to local practicalities.**    Many participants thought that time expectations, for example the number and the length of sessions, may need adaptation for any caregiver intervention. They thought that changes in the intervention's delivery model might also be helpful. They gave examples from CST: home visits form a core part of the intervention in between sessions. However, in some settings home visits were not possible because of security or logistical difficulties. Therefore, the intervention facilitators had to find another venue, for example a community centre.

*PCP3, clinician, Americas*

*"We might have some limitations to get into the houses, it's a very poor area, it might be like conflict places, so we don't know yet, it might happen that we have to do the home visits in another place, in a centre that we rent for them."*

**Settings with competing services.**    Participants discussed that in settings where a range of interventions is already available for caregivers, providers might be in competition with one another. They thought that caregivers may also be confused about whose advice to take first, thereby making the implementation of a new service challenging. They added that those facilitating the interventions can benefit from having experience with more than one programme and this experience can contribute to a successful intervention delivery.

*PCP20, clinician, Western Pacific*

*"We have very cheap national health insurance. And most of the early intervention is covered by this insurance. And actually in the urban [areas] there's a lot of resources of the early*

*intervention. So [. . .] these parents will have more ability to find resources for the children. So in our first six groups, most of the parents they have a lot of interventions for their child. Not only CST. So they were comparing this intervention with other interventions like ABA, the therapy ABA taught me we have to do like this and the facilitator in CST taught me I have to do this, and the other therapist taught me to do something, so they are confusing*"

One solution that participants proposed could be building the new programme as foundational level service, while competitors offer more specialised care. Another possibility is building a referral network around the intervention to provide more comprehensive support to caregivers.

## Understanding the reality of caregivers

When discussing experiences with supporting caregivers, an overarching concern of most participants was the socio-economic context of families both from the caregivers' and from the intervention providers' perspective. Many thought that addressing this should be prioritised over culture in the adaptation process, since it has an immediate impact on the accessibility, availability, and affordability of a programme. Participants put forward that in low resource settings there are no or few services available: with lengthy waiting lists in public health or costly private care, a lack of trained professionals and very few educational opportunities to specialise in DDs. They indicated that in such cases, caregivers may accept any support without questioning programme effectiveness. Informants discussed a range of challenges caregivers may face: they may not have a stable job, have limited financial resources and lack childcare.
*PCP3, clinician, Americas*

*"The reality of the population that we were working [with]. . .sometimes it happened that the mother started coming to the sessions and then all of a sudden get the opportunity to get a job so couldn't continue to come and then the [other] parent start coming. . .and then the big brother came, so I mean it's not the ideal thing but it was what was possible."*

**Education and literacy of caregivers.** Participants mentioned that the educational level of caregivers should be taken into account when adaptations are made, to allow for participation of caregivers who have received limited or no education. They said that while in lower resource settings caregivers may not be familiar with effective interventions that exist; in higher income, urban settings parents may already be knowledgeable about evidence-based services available. Many participants raised that intervention materials should be written in an accessible, simple language. Technical terms, such as 'joint engagement' or 'home routines' should be explained in a way that is easy to understand for a lay audience. Particularly participants from the Americas region mentioned the difficulty of managing illiteracy. To support caregivers with low literacy, they suggested that written materials could be adapted to a format accessible without reading.
*PCP13, clinician, Americas*

*"The thing I was wondering was about the illiteracy of the people, I think it's the most difficult part of the project because the CST programme has a really good material and if people don't read they won't be able to get in touch with this material and we have a lot of illiteracy in our country."*

**The location and venue of the programme.** Informants mentioned that caregivers may live in remote areas where public transportation is scarce. They added that caregivers may feel uncomfortable travelling to or attending sessions in certain institutions or venues.

*PCP129, caregiver, Western Pacific*

*"Where your targets are the indigenous families, will they be able to get the help that they need? Many people won't go to our public hospitals. . .So we start from the smallest unit. . .they have social workers in there, then if this family, if they see a social worker, they know they need help."*

Participants discussed that transportation options and affordability of transportation should both be considered. In rural areas families often have to walk long distances to access services. Those working in remote zones explained strategies that help caregivers engage with interventions. Community health workers can spread the word about available programmes in remote communities, or a reimbursement of travel costs can be offered to remove financial barriers of attendance. Participants expressed that in low resource settings the choice of intervention is often different from a resource-rich context: it is likely based on what is freely accessible, scalable, can be delivered at a minimal cost and is easily manageable by non-specialists.

**Supporting families in poverty.** Many informants thought that caregiver-mediated interventions can be an effective way of providing services in low-resource contexts and settings where few specialists are available. When caregivers living in poverty do have access to interventions, they may still face challenges: for example in arranging childcare. Families living in poverty in certain settings receive governmental benefits because of the child with an DD. Some families opt out of attending the intervention because they fear losing this financial aid from the government.

*PCP18, clinician, Africa*

*"It's hard for them [caregivers] to find someone who would sit in with their child and come for the assessment. So having to look for someone, whom they would pay, so they can look after the child for the time that they are in for the session, it was difficult for some of them, not every person was willing to assist in that nature."*

## Challenges of sustaining an intervention

All informants discussed the importance of having caregiver interventions that are sustainable in the long run.

**Supporting families in the context of competing priorities.** An overarching question was how decisions are made about supporting persons with DDs when there are competing areas of need. According to the participants, priority setting can be observed within families and at a governmental level. Informants explained that caregivers find different ways of meeting the needs of their child with a DD and those of other family members. Some choose to look after the child with a DD over everything else, while others prioritised generating income for the family above one-to-one care for their child with a DD. Other informants highlighted the impact that social determinants have on how families set priorities, for example in conflict-affected zones and refugee settings.

*PCP19, researcher, Eastern Mediterranean Region*

*"In the refugee setting, you live in a tent and. . . they have on average five kids per family, in addition to that you have the economical stressors and of course the life hazards that come*

*with living in a tent and sharing it with other families. So the caregiver then would be in a different state of mind to focus their energy and attention on doing these practices at home."*

Informants explained that on a governmental level, DDs often get overlooked when health priorities are set. Participants from international organisations added that this can be because of a lack of awareness of DDs or the fact that the responsibility to manage DDs is split across different sectors, including health, social care, and education. Preventable diseases and conditions responsible for child mortality might be prioritised in health services over DDs, especially in LMICs. Other participants added that many policymakers are prioritising investment in areas where visible change can more easily be observed.

**The role of funding in sustaining a programme.** Informants thought that a consequence of the low priority afforded to DDs was that funding would rarely be allocated for interventions for families with children with DDs. Participants from all countries experienced difficulties getting funding for any work related to DDs and for mental health-related aspects of public health services. An example is a setting where funding for CST was cut due to changes in the funding organisation's priorities and hence all support for the local caregivers stopped. Funding constraints can result in difficulties in forming partnerships and issues in compensating professionals for their efforts. As a result, professionals might end up working on a voluntary basis. Informants reported trying to avoid reliance on volunteerism due to the risk of losing staff if they were not adequately paid to do their job.

*PCP126, caregiver, Americas*

*"We have this problem of the payment of the professionals who carry this out in the public hospitals. So many times, these professionals in the end make it for free. So it's not because they left the programme or left the hospital because the programme was not good, they left probably before because. . .they had to buy their grocery too."*

Many participants thought that the involvement of government services was crucial to maintaining the intervention, with facilitators being community-based, contracted by the government. Others mentioned that teaching the intervention as part of the pre-service training of students in relevant fields could help to build a sustainable system.

When discussing funding, participants thought that there were at least two considerations to be made: sustainability and the level of influence the funding body would have over how the programme was run. Finally, most participants agreed that stakeholders' and the funder's interests can directly impact how the intervention is organised. Some informants mentioned that the funder may have different expectations or outline who should be involved in a programme thereby setting conditions. Others added that in fast-changing political systems a change in government may result in big changes to support for programmes.

*PCP9, researcher, Western Pacific*

*"Different people have different opinions on which organisation to give how many quotas [the number of CST master trainers (MTs) to be trained], because the university does play a part and the funder does play a part and they do have some kind of a. . .what I would say conflict, different opinions. Some people wanted to have a good collaborative relationship with these NGOs, so giving the MT quotas to these NGOs. . .but the funder's point of view, they wanted us to give the quotas to other NGOs."*

**Coordinating long-term stakeholder collaborations.** Informants raised that developing and then scaling up a caregiver intervention requires the collaboration of a range of

stakeholders, including advocacy groups, clinicians, funders, researchers, caregivers, and various ministries. A challenge that participants reported was coordinating all these groups: it may be difficult to keep communication up to date; resources for the intervention may be inadequate; and the preferred communication channel may differ per stakeholder group. Some participants added that coordination may be difficult when stakeholders work in fragmented organisations, or when an organisation has priorities different from the goals of the intervention.

*PCP126, caregiver, Americas*

*"The [CST] pilot started in the psychiatric hospital. We were looking for different stakeholders, but that was too vast and ambitious. The ministry of education, of health, of social security, of civil society, the municipality, and there were so many expectations that finally, the hospital decided to do it on their own. Without the rest of the stakeholders. Because everybody talked a lot and nothing was reached."*

Many participants said that the initiative to adapt and implement the intervention was taken by one driven individual with strong personal motivation. Some informants thought that personal motivation to take part helped managing coordination. However, informants emphasised that the programme should not rely on only one person's motivation to keep the intervention going in the long run, as it is not sustainable. A suggested example to improve sustainability was to let a governmental body take ownership of implementing the intervention.

Finally, many participants argued that a solution to coordination challenges is to use participatory approaches in the adaptation and implementation and to involve everyone in the decision-making. Informants involved in advocacy added that caregivers can contribute their expertise and experience on DDs.

## Discussion

In this study we investigated the role that context plays in adapting and implementing caregiver interventions for families raising children with DDs across settings. In Phase 1 we looked specifically at the WHO Caregiver Skills Training programme, while in Phase 2 we expanded the focus to include caregiver interventions more generally. Four main themes were developed. While some participants thought that intervention strategies and psychoeducational messages for caregivers apply universally, others pointed out the need for local adaptations. Many believed that broader contextual and structural factors played an even more important role than culture by itself. Informants thought that socio-economic factors, including poverty, impact whether caregivers have access to services and how they can commit and engage with them. Participants thought that longer term perspectives should be taken into account as part of the adaptation process. Many wondered about the ways in which an intervention can sustain and benefit caregivers, even in contexts where DDs might not always be prioritised.

The participants in our study emphasised that contextual adaptations are relevant to make sure that an intervention meets local needs. This finding resonates with previous research highlighting the importance of addressing caregiver needs locally [58]. In existing literature cultural sensitivity is often emphasised as a key to successful adaptation over and beyond other contextual factors [59]. Our informants had different views regarding the importance of culture in adaptations. Many thought that other contextual factors, such as the location and socio-economic environment, should be prioritised since they directly impact the feasibility and acceptability of the intervention. Participants mentioned a range of elements that may

require adaptation, such as translations, changes in illustrations, using online tools, and adapting psychological concepts, as shown in previous studies as well [60]. Taking stakeholder views on adaptations into account can contribute to a richer conceptualisation of a context [36], allowing for the intervention to be acceptable, accessible, feasible and sustainable.

Participants reported that some of the technical terms describing core ingredients of interventions are hard to translate, difficult for caregivers to understand, or rely on alien conceptualisations of the role of caregivers. An example they mentioned from the CST programme was the concept of joint engagement. A core goal of the CST is to increase the time that the caregiver and the child spend in joint engagement experiences: sharing attention with a partner on a joint activity. In CST the focus is on building these joint engagement experiences by transforming common activities such as play and home activities into "routines". Cultural beliefs regarding how and when to involve children in everyday contexts may affect the understanding of key intervention concepts. For example in Italy the adaptation of CST required careful wording and addition of examples to define the terms 'activity', 'routine' and 'engagement' [61]. Our participants explained that caregivers and children may engage in everyday activities differently across settings. Given that play routines are a primary context for the development of joint engagement, it is possible that cultural differences in the frequency and attitude to play may have exacerbated difficulties in understanding its concept. Moreover, existing literature shows that the idea that children only play and go to school is a largely Western concept [62]. Children in lower resource settings tend to integrate work, play, and school, often taking on adult responsibilities, impacting the caregiver-child relationship [62]. This means that different strategies may be needed across settings for caregivers to understand and implement join attention with their children. The challenge with the adaptation of such strategies is when they affect the active ingredients of the interventions' efficacy. For example, a previous autism intervention study in the USA suggested up to 69% of the intervention effect on language improvement may be mediated by joint engagement [63]. Using local strategies to achieve joint engagement may be variably effective. To ensure that active ingredients continue to work as intended across settings, it is important to include caregiver priorities regarding parenting when delivering interventions. This should involve ensuring that relevant terminology is explained through examples that are culturally and contextually relevant. Meanwhile, the question also arises as to whether certain psychological concepts common to early interventions for children with DD are perhaps not culturally universal. Examples include *tension* being a specific idiom of distress in India [64] or *respect* a specific Latin American value of parenting [65]. Exploring and using local understandings of health, child development and parenting can be helpful in overcoming such issues [66]. Further research into how best to keep the core elements of an intervention consistent across contexts, while communicating them using local ways of understanding would be helpful.

Addressing poverty and socio-economic factors have previously been highlighted as crucial components when adapting caregiver interventions [67] or responding to caregiver needs [58]. Having a family member with a DD is associated with lower household income and quality of life of families [68]; furthermore, poverty is associated with reduced access to health care [69]. The third theme considered how the socio-economic context and poverty may impact on the adaptation and implementation process. Informants discussed aspects of location, the education and literacy of caregivers and their income level. They emphasised how such factors affect participation in, and engagement with, the programme. For example, in certain low-income settings caregivers may not be able to find and pay for childcare for the duration of an intervention. Informants raised other points showcasing how socio-economic factors can pose a systemic barrier to accessing support. Caregivers may not be informed about DDs (believing that DDs can be cured) and they may not have accurate information about their rights (when

would the receipt of financial benefits stop). These difficulties could be likely to arise because of complex societal processes posing a barrier on those with less social power and capital, as suggested by intersectionality scholars studying social determinants of health [70]. The examples also suggest that the socio-economic environment and the income level of beneficiaries should receive more attention when caregiver interventions are adapted across settings, both from a research and a policy perspective [71, 72].

Participants thought that sustainable financing and long-term collaborations across stakeholder groups can help the intervention become sustainable in the longer term. Previous literature suggests that integrating an intervention to public services means that the role of programme implementers decreases. Instead, public service workers start playing a decision-making role. They become the ones making sure that the programme is sustainably funded, has ongoing demand, and is being monitored and evaluated [73]. Our findings add to this framework by outlining challenges that teams adapting and implementing caregiver interventions may face: for example, how to manage the intervention in an environment where DDs are not considered as a health priority. A possible solution that informants mentioned is if the intervention forms part of the pre-service training for students in relevant fields. There is existing evidence showing that this method can be an efficient way of training intervention facilitators, for example as part of the WHO Mental Health Gap Action Programme [74].

Financial sustainability was of concern for participants. Even though a programme is based on training non-specialists and using task-sharing methods, maintaining the programme comes with certain costs and access to funding might already be biased against vulnerable groups. Additionally, participants emphasised that not offering a fair pay to intervention facilitators can result in attrition. This has been shown in other studies too: many public service initiatives rely on unpaid volunteer work, especially in LMICs [8]. While the WHO recommends a fair wage for health workers, a general lack of funding prevails, hence volunteer work is common [75] which might be exploitative [76].

## Implications

Barriers such as poverty and lack of resources exist both in terms of providing interventions and being able to utilise them as a beneficiary. Previous research has highlighted the need for multiple interventions in public health, poverty eradication and international development in combination to remove these access barriers [77]. Our results point to a similar direction: while cultural factors are indeed important to consider, the broader socio-economic context may have an even more relevant role. Example strategies to address challenges set by the socio-economic context may include collaboration with poverty eradication initiatives, providing care for families living in remote areas or caregivers with low levels of education [78, 79].

Our findings show the rich variety of changes that can occur when adapting a programme and therefore we endorse the following suggestions. Keeping track of why and how adaptations are made can reinforce the evidence-base of caregiver interventions when they are adapted across settings. Well-documented adaptations can be informative for settings with a similar context, therefore facilitating the implementation of an intervention [80]. This is a strategy that programmes like the WHO CST have already been using by providing an adaptation guide and record sheet to systematically document how the programme is adapted [28]. Collaborations between service providers, caregiver and grassroots organisations and policy makers in the adaptation process can help address cultural differences and can be a starting point of a longer-term stakeholder engagement. Lastly, throughout the adaptation funding models should be explored that allow interventions to be sustainable and offer a fair compensation to facilitators.

Further research would be helpful to understand whether the core elements of caregiver-mediated interventions for DDs are culturally universal. If they are, it would then be important to investigate how contextual factors, including cultural differences in caregiver-child engagement may impact their local implementation. When new strategies are developed to explain and implement psychological concepts across settings, rigorous and systematic evaluation is needed to understand whether they remain evidence-based.

## Limitations

A limitation of this study is that all participants were stakeholders directly involved in adapting or implementing the CST programme. Including voices of those working with different interventions, perhaps also some who may deliberately have chosen a different intervention because they are critical of the CST programme, may have provided more diverse information. As the FGDs took place during a WHO CST meeting, participants may not have felt comfortable sharing negative, or differing perspectives about the CST programme. A further limitation is that the perspectives of stakeholders who leave the CST or stop running caregiver interventions were underrepresented: there was only one stakeholder interviewed who left working on the CST programme. The study is also limited in terms of its geographical representation. Most of the participants came from the Americas, and regions such as Africa or Europe are not well-represented. Interviews took place online: this meant that potential interviewees coming from contexts lacking stable internet connection were not able to participate. Beside the interviews in English, two interviews in Spanish were conducted, transcribed, and translated by ZS who is neither a native Spanish nor a native English speaker. This may have lost some of the subtlety of idiomatic use of language. English was not the first language of most participants, and this likely impacted the clarity and fluency of the interviews.

## Conclusion

In this study we investigated the role of context in adapting and implementing caregiver interventions for DDs. While cultural adaptation is important, other contextual factors such as the intervention location, and the education and literacy of caregivers may have an even stronger impact on intervention uptake and effectiveness. The socio-economic context of caregivers and poverty strongly influence service access and engagement with interventions. Challenges to integrating an intervention to public services include competing health priorities, lack of funding and long-term management of relationships with stakeholders.

## Supporting information

**S1 File. The WHO approach to adapting the CST programme.**
(DOCX)

**S2 File. Focus group discussion topic guide.**
(DOCX)

**S3 File. Individual interview guideline.**
(DOCX)

**S4 File. Interview guide for WHO and autism speaks representatives.**
(DOCX)

**S1 Table. Exemplar quotes.**
(DOCX)

## Acknowledgments

We would like to thank the whole WHO CST Team for their support with this study. The WHO CST Team includes (in alphabetical order): Felicity L. Brown (War Child Holland and University of Amsterdam, Netherlands), Laura Pacione (University of Toronto, Canada), Erica Salomone (University of Milano-Bicocca, Italy), Chiara Servili (WHO), Stephanie Shire (University of Oregon, USA).

## Author Contributions

**Conceptualization:** Zsofia Szlamka, Charlotte Hanlon, Rosa A. Hoekstra.

**Data curation:** Zsofia Szlamka.

**Formal analysis:** Zsofia Szlamka, Bethlehem Tekola.

**Funding acquisition:** Zsofia Szlamka.

**Methodology:** Zsofia Szlamka.

**Project administration:** Zsofia Szlamka.

**Resources:** Zsofia Szlamka.

**Software:** Zsofia Szlamka.

**Supervision:** Charlotte Hanlon, Rosa A. Hoekstra.

**Writing – original draft:** Zsofia Szlamka.

**Writing – review & editing:** Zsofia Szlamka, Charlotte Hanlon, Bethlehem Tekola, Laura Pacione, Erica Salomone, Chiara Servili, Rosa A. Hoekstra.

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
