## [Decision Letter · Decision Letter 0]

4 Mar 2022

PONE-D-21-29533Perspectives on the role of context in adapting caregiver interventions for families raising children with neurodevelopmental disordersPLOS ONE

Dear Dr. Szlamka,

Thank you for submitting your manuscript to PLOS ONE. I have received comments from two experts in the field and reviewed your manuscript personally. I agree with the reviewers that this is a well written manuscript and provides a useful contribution to the field. As you will see, the reviewers had several comments. In general, I think the concerns are relatively minor, but will need to be addressed prior to publication.

We look forward to receiving your revised manuscript.

Kind regards,

Eric J. Moody, Ph.D.

Academic Editor

PLOS ONE

Journal Requirements:

2. During our internal checks, the in-house editorial staff noted that you conducted  the phase 1 of your research in another country. Please check the relevant national regulations and laws applying to foreign researchers and state whether you obtained the required permits and approvals. Please address this in your ethics statement in both the manuscript and submission information. In addition, please ensure that you have suitably acknowledged the contributions of any local collaborators involved in this work in your authorship list and/or Acknowledgements. Authorship criteria is based on the International Committee of Medical Journal Editors (ICMJE) Uniform Requirements for Manuscripts Submitted to Biomedical Journals - for further information please see here: https://journals.plos.org/plosone/s/authorship

Reviewers' comments:

Reviewer's Responses to Questions

**Comments to the Author**

1. Is the manuscript technically sound, and do the data support the conclusions?

Reviewer #1: Yes

Reviewer #2: Yes

2. Has the statistical analysis been performed appropriately and rigorously? 

Reviewer #1: N/A

Reviewer #2: N/A

3. Have the authors made all data underlying the findings in their manuscript fully available?

Reviewer #1: Yes

Reviewer #2: Yes

4. Is the manuscript presented in an intelligible fashion and written in standard English?

Reviewer #1: Yes

Reviewer #2: Yes

5. Review Comments to the Author

Reviewer #1: This is a well written article which I think will be very valuable in the field. In general it is well presented so I have only a few comments to make, such as additions to the limitations paragraph.

Small points:

I think the title could be clearer: the word 'perspectives' is unnecessary (as that is the method), and probably the word 'role'. There could instead be emphasis on the necessity of appropriate contextual adaptation.

There are a few places where there are word or grammatical oddities: e.g. page 6, participants paragraph, line 6 'again'; page 17, second paragraph, line 3, delete 'of'.

Table 1 - I think it would be better to combine the two groups.

Page 4: 'inner and outer barriers' requires more explanation.

Page 26: first paragraph. The sentence needs to be unpacked more. Start a new sentence when describing the understanding of 'tension' in mental health, and so on.

In Limitations, the authors could add that only one person coded, apparently with no collaborative discussion or checks on ratings (or add to Methods if there were). Also the 23 interviews in English were mostly not in the interviewees' first language, which also is likely to limit clarity and fluency. Finally, there was not an opportunity to have participant reflection on the themes derived.

More substantive comments:

1) It read well when participants' solutions to issues were also included in the presentation of results.

2) On page 4, the content of WHO CST adaptation guide is mentioned briefly. Given the findings of this study, did it require amendments? For example, the quote on page 24 comments on the difficulty of following the process advised of including a range of stakeholders.

3) Perhaps on page 5, it might be helpful to give an example of cultural adaptation - e.g. using the term 'ubuntu' in Africa to ground early intervention (e.g. see review by Smythe et al International Health, May 2021).

4) On page 14, under Psychological concepts, 'joint engagement' is mentioned as culturally difficult. I'm not sure it is cultural, as in my experience it takes time in Western settings also for many parents to grasp what is meant. However the rest of this paragraph is insufficient as the text only mentions 'play', and JE is more than play. The quote does not really illustrate the issue clearly either, as it is about authoritarian parenting style. I suggest changing the example (since JE comes up again later - see my next point) or explain it more fully here, perhaps with an example of how it has been successfully explained in one of the settings.

5) On page 25/26, there could be further discussion of the issue raised in the introduction of when an adaptation goes outside the evidence base of the intervention (as opposed to utilising local concepts to create an explanation of an unfamiliar one). In the WHO CST approach, the inclusion of caregiver-child interaction strategies leading to 'joint engagement' is not really optional. It could be helpful to cite papers looking at the active elements of this kind of approach to intervention (e.g. Gulsrud et al Journal of Child Psychology and Psychiatry, 2016; Shih et al JCPP, 2021).

Reviewer #2: Thank you for the opportunity to review this manuscript, entitled, ‘Perspectives on the role of context in adapting caregiver interventions for families raising children with neurodevelopmental disorders’.

This is, in general, a well conceptualised and written paper, methodologically sound and with results that are relevant and meaningful. The discussion is carefully considered and integrates appropriate literature. As a result, I have relatively few comments and suggestions and would like to congratulate the authors on their work in this important area.

Introduction

The introduction is concise and addresses the salient areas. I did wonder if it would be useful to add some pieces regarding the risks or challenges of task sharing, particularly in LMIC (which could then link in with the challenges of funding and sustainability in the discussion section later). This aside, I don’t have any other specific suggestions for changes.

Methods

I would have liked to know how many people attended the technical consultation meeting in Xiamen, China – 15 accepted the invitation to participate in the focus groups, leaving how many who did not?

Were there cases where participants were both a researcher and a clinician? If so, how was it decided where they were assigned in the table?

It would be useful to know a little more about the participants and their background, specifically their experience with the CST, e.g. were most of them in the starting phase, how many had completed the adaptation process etc. How many worked in NGO’s vs public service etc. While there was an excellent range, if there was a big proportion working in NGO’s for example that would be important to know.

The researchers positionality was thoughtfully described, I would suggest it be important to describe their own background (i.e ZS and RAH), given it’s relevance and potential for bias. E.g. are both researchers or also clinicians, do they hold more than one role and was this shared with the group?

Results

The results are well described and concisely articulated. The use of quotes is appropriate and adds to the section.

Perhaps I missed it, but I think it would be helpful to have a brief description/explanation that data from phase 1 and 2 were amalgamated in the results section.

When sharing quotes from participants, individuals were given different descriptions – e.g. ‘clinician’ or ‘caregiver and advocate’, others had more detailed descriptions, e.g. ‘trainer of master trainers’. I would suggest, if possible, more consistency, perhaps one of clinician/researcher/caregiver and then an additional description of the participant (if possible)?

Discussion

The discussion was considered, included relevant literature and outlined some of the complexity of this subject area. The authors at times chose to reflect opposing views expressed by the participants which added to the richness of the section. The key sections of the results are elaborated on and examined.

Although the implications section raises salient points, I wonder if there could be a little more on reflections for future research in this area. Given how challenging it is to capture the role of context in adapting and implementing caregiver interventions, providing some guidance in the area would be useful.

Page 25, last paragraph, line 5, perhaps this could be rewritten for clarity.

Limitations

I think a line on the ways in which not having stakeholders who left CST participate in the study is a limitation, would be useful.

Is a further limitation of the study that it only included stakeholders directly involved in CST? I would think that gathering information from those not involved in the programme (and outside the structure and framework of WHO) may gather further, and diverse information. Given this was a CST meeting, I also wonder if, as was alluded to, there may not have been some bias, that participants may have felt a need to be largely positive about the ways in which CST was managing adaptations and implementation (e.g. not emphasising those areas it was not managing so well).

References

I would suggest a look over the references for consistency, I noted for example #’s 8, 13, 58 and 62

6. PLOS authors have the option to publish the peer review history of their article (what does this mean?). If published, this will include your full peer review and any attached files.

Reviewer #1: No

Reviewer #2: **Yes: **John-Joe Dawson-Squibb

---

## [Author Response · Author response to Decision Letter 0]

13 Apr 2022

09.04.2022 

Dear Dr. Eric J. Moody, 

Re: Response to reviewers of Manuscript [PONE-D-21-29533] - [EMID:7dd3a3cb4104695a] 

Title: Exploring contextual adaptations in caregiver interventions for families raising children with developmental disabilities 

Thank you for your invitation to revise and resubmit our manuscript following the helpful comments from two reviewers. Below we have responded in detail to each of the reviewers’ comments. Reviewer comments are in italicised text and our responses are inserted after each comment in plain text. The revisions are also highlighted in the manuscript text itself using coloured text. 

Reviewer #1: This is a well written article which I think will be very valuable in the field. In general it is well presented so I have only a few comments to make, such as additions to the limitations paragraph. 

Thank you for reviewing our manuscript and providing helpful comments. We highlighted in yellow our additions and edits addressing your points in the manuscript. 

Small points: 

Comment 1. I think the title could be clearer: the word 'perspectives' is unnecessary (as that is the method), and probably the word 'role'. There could instead be emphasis on the necessity of appropriate contextual adaptation. 

We considered your point and revised the title and subtitle accordingly. It now reads as follows: 

“Exploring contextual adaptations in caregiver interventions for families raising children with developmental disabilities” 

Comment 2. There are a few places where there are word or grammatical oddities: e.g. page 6, participants paragraph, line 6 'again'; page 17, second paragraph, line 3, delete 'of'. 

We reviewed the manuscript for word and grammatical errors and made minor corrections through the manuscript, highlighted in blue. 

Comment 3. Table 1 - I think it would be better to combine the two groups. 

We combined the two groups in Table 1 and it now looks as follows: 

Region 

Overall 

Clinicians 

Caregivers 

Researchers 

Africa 

1 

n/a 

n/a 

1 

Americas: 

North America 

1 

n/a 

n/a 

1 

Central and South America 

8 

5 

3 

n/a 

South-East Asia  

1 

1 

n/a 

n/a 

Europe 

0 

n/a 

n/a 

n/a 

Eastern Mediterranean Region 

0 

n/a 

n/a 

n/a 

Western Pacific  

4 

2 

1 

1 

Comment 4. Page 4: 'inner and outer barriers' requires more explanation. 

We revised this paragraph to simplify the way in which we explain the various understandings and definitions of context in the adaptation literature. We removed the sentence about inner and outer barriers. The paragraph now reads as follows: 

In the existing literature there is no clear agreement as to what context means. Some studies consider contextual changes to be part of the implementation and not the adaptation process (30,31). Many scholars agree that contextual adaptation goes beyond the geographic setting (32,33). Previous studies mentioned examples such as activity types in the intervention and broader societal discourses about the adaptation and implementation process (28,34); organisational and community norms and attitudes (35); or the physical and environmental surroundings of the intervention, socio-economic factors and fidelity (36). Meanwhile in the adaptation literature there has been a strong focus on cultural adaptations as compared to other contextual factors (26,27,37–42). 

Comment 5. Page 26: first paragraph. The sentence needs to be unpacked more. Start a new sentence when describing the understanding of 'tension' in mental health, and so on. 

We unpacked the first sentence of the first paragraph and it now reads as follows: 

The participants in our study emphasised that contextual adaptations are relevant to make sure that an intervention meets local needs. This finding resonates with previous research highlighting the importance of addressing caregiver needs locally (53). 

In the second paragraph we started a new sentence when we mention the understanding of tension in mental health, and it now reads as: 

Meanwhile, this difficulty poses the question as to whether certain psychological concepts common to early interventions for children with DD are perhaps not culturally universal. Examples include tension being a specific idiom of distress in India (56) or respect a specific Latin American value of parenting (57). 

Comment 6. In Limitations, the authors could add that only one person coded, apparently with no collaborative discussion or checks on ratings (or add to Methods if there were). Also the 23 interviews in English were mostly not in the interviewees' first language, which also is likely to limit clarity and fluency. Finally, there was not an opportunity to have participant reflection on the themes derived. 

Thanks for this helpful comment: it made us realise that it was reflexive thematic analysis that we used. We also clarified what we meant by the iterative analysis of data. The full dataset was coded by ZS and the first three individual interviews and the two FGDs were also coded by BT. Co-authors iteratively discussed the development of the codebook and themes as part of thematic analysis. We now clarified this in the Analysis section, which reads as follows: 

Data were anonymised and analysed using reflexive thematic analysis with the qualitative data management software NVivo 12 (50). 

Coding started upon completion of the FGDs. The first coding of FGDs was done by ZS and in discussion between ZS, RAH and CH, allowing for investigator triangulation (47). Upon completion of the individual interviews, data from Phase 1 and Phase 2 were merged and the inductive analysis continued using the whole dataset. The first three individual interviews and the two FGDs were also coded by a second coder (BT), who is an Ethiopian Social Scientist with has extensive experience in qualitative research and who has worked previously on the adaptation of CST in Ethiopia (75). The two coders then shared their codebooks, discussed preliminary themes they developed and agreed on a codebook. ZS coded the remainder of the interviews and further refined the themes and subthemes based on discussion with RAH and CH. 

We added to the Limitations that English was not the first language of interviewees and it now reads as: 

English was not the first language of most participants, and this likely impacted the clarity and fluency of the interviews. 

We sent a summary document of the preliminary themes after the FGDs to participants for feedback and reflection. We added a Patient and participant involvement paragraph on page 10 just before the Analysis and it now reads as follows: 

Patient and participant involvement 

A document summarising the preliminary results from the FGDs were sent to participants for feedback and reflection. Participant feedback then informed the development of individual interviews. 

Comment 7. More substantive comments: 

It read well when participants' solutions to issues were also included in the presentation of results. 

Thank you. 

Comment 8. On page 4, the content of WHO CST adaptation guide is mentioned briefly. Given the findings of this study, did it require amendments? For example, the quote on page 24 comments on the difficulty of following the process advised of including a range of stakeholders. 

Phase 1 of this study (with focus group discussions) looked at the adaptation process through the lens of the WHO CST specifically. However, in Phase 2 we investigated stakeholder experiences with adaptations in general, including, but not limited to the CST. Therefore, not all the results were directly applicable to the CST. There were no specific adjustments made to the WHO CST adaptation guide. However, this work validated WHO’s approach to maximising feasibility of the CST. This approach emphasized the importance of contextual adaptation, especially consideration of caregivers’ socio-economic context, including poverty. 

Comment 9. Perhaps on page 5, it might be helpful to give an example of cultural adaptation - e.g. using the term 'ubuntu' in Africa to ground early intervention (e.g. see review by Smythe et al International Health, May 2021). 

We added this example to the text and it reads as follows: 

An example to this cultural focus is using the term ‘ubuntu’ to ground early interventions for DDs in Africa (75). 

Comment 10. On page 14, under Psychological concepts, 'joint engagement' is mentioned as culturally difficult. I'm not sure it is cultural, as in my experience it takes time in Western settings also for many parents to grasp what is meant. However the rest of this paragraph is insufficient as the text only mentions 'play', and JE is more than play. The quote does not really illustrate the issue clearly either, as it is about authoritarian parenting style. I suggest changing the example (since JE comes up again later - see my next point) or explain it more fully here, perhaps with an example of how it has been successfully explained in one of the settings. 

Thank you for this comment. We revised the relevant paragraph which is now on Page 16. We clarified that it is play and how children and caregiver engage with one another in a certain context that can differ across settings. It is these differences in engagement that mean difficulties in explaining psychological concepts such as joint engagement. It now reads as follows: 

Informants raised that some psychological concepts core to the intervention can be difficult to translate across settings and many gave the example of the term ‘joint engagement’. Many thought that the concept of joint engagement was difficult to understand for caregivers. Some believed that it was because of cultural differences in play. They raised that engagement through play in which they caregivers and children act as equal partners does not come naturally to caregivers in their context. Such differences can mean that some intervention strategies are not easy to understand for caregivers and therefore harder to implement. 

Comment 11. On page 25/26, there could be further discussion of the issue raised in the introduction of when an adaptation goes outside the evidence base of the intervention (as opposed to utilising local concepts to create an explanation of an unfamiliar one). In the WHO CST approach, the inclusion of caregiver-child interaction strategies leading to 'joint engagement' is not really optional. It could be helpful to cite papers looking at the active elements of this kind of approach to intervention (e.g. Gulsrud et al Journal of Child Psychology and Psychiatry, 2016; Shih et al JCPP, 2021). 

Thank you for this helpful comment. We added various points and the references suggested to the relevant paragraph of the Discussion on page 27, including that the evidence-base may be lost by dropping core components of an intervention for locally used concepts. The edited text reads as follows: 

Participants reported that some of the technical terms describing core ingredients of interventions are hard to translate, difficult for caregivers to understand, or rely on alien conceptualisations of the role of caregivers. An example they mentioned from the CST programme was the concept of joint engagement. A core goal of the CST is to increase the time that the caregiver and the child spend in joint engagement experiences: sharing attention with a partner on a joint activity. In CST the focus is on building these joint engagement experiences by transforming common activities such as play and home activities into “routines”. Cultural beliefs regarding how and when to involve children in everyday contexts may affect the understanding of key intervention concepts. For example, in Italy the adaptation of CST required careful wording and addition of examples to define the terms ‘activity’, ‘routine’ and ‘engagement’ (80). Our participants explained that caregivers and children may engage in everyday activities differently across settings. Given that play routines are a primary context for the development of joint engagement, it is possible that cultural differences in the frequency and attitude to play may have exacerbated difficulties in understanding this concept. Moreover, existing literature shows that the idea that children only play and go to school is a largely Western concept (79). Children in lower resource settings tend to integrate work, play, and school, often taking on adult responsibilities, impacting the caregiver-child relationship (79). This means that different strategies may be needed across settings for caregivers to understand and implement join attention with their children. The challenge with the adaptation of such strategies is when they affect the active ingredients of the interventions’ efficacy. For example, a previous autism intervention study in the USA suggested up to 69% of the intervention effect on language improvement may be mediated by joined engagement (74). Using local strategies to achieve joint engagement may be variably effective. To ensure that active ingredients continue to work as intended across settings, it is important to include caregiver priorities regarding parenting when delivering interventions. This should involve ensuring that relevant terminology is explained through examples that are culturally and contextually relevant. Meanwhile, the question also arises as to whether certain psychological concepts common to early interventions for children with DD are perhaps not culturally universal. Examples include tension being a specific idiom of distress in India (56) or respect a specific Latin American value of parenting (57). Exploring and using local understandings of health, child development and parenting can be helpful in overcoming such issues (58). Further research into how best to keep the core elements of an intervention consistent across contexts, while communicating them using local ways of understanding would be helpful. 

Reviewer #2: Thank you for the opportunity to review this manuscript, entitled, ‘Perspectives on the role of context in adapting caregiver interventions for families raising children with developmental disabilities. 

This is, in general, a well conceptualised and written paper, methodologically sound and with results that are relevant and meaningful. The discussion is carefully considered and integrates appropriate literature. As a result, I have relatively few comments and suggestions and would like to congratulate the authors on their work in this important area. 

Thank you for reviewing our manuscript and for providing helpful comments to improve the work even further. We highlighted with yellow our additions and edits addressing your points in the manuscript in yellow. 

Comment 1. Introduction 

The introduction is concise and addresses the salient areas. I did wonder if it would be useful to add some pieces regarding the risks or challenges of task sharing, particularly in LMIC (which could then link in with the challenges of funding and sustainability in the discussion section later). This aside, I don’t have any other specific suggestions for changes. 

We added the following points to the introduction about task sharing: 

Task sharing is suggested to help increase access to care from high- to low-income settings (77). However, critiques proposed that task sharing may not always be acceptable and feasible in lower resource contexts: non-specialists may not feel competent to deliver certain tasks for example (78). 

Comment 2. Methods 

I would have liked to know how many people attended the technical consultation meeting in Xiamen, China – 15 accepted the invitation to participate in the focus groups, leaving how many who did not? 

Overall, 42 participants from 20 countries were present in the technical consultation meeting. Participants in our study were drawn from six of the countries for the FGDs. We clarified this in the Methods section in the paragraph about Phase 1 on page 6: 

Two focus group discussions (FGDs) were conducted during a technical consultation meeting of the WHO CST in Xiamen, China on 8-9th November 2018, an event attended by 42 participants from twenty country teams adapting and implementing CST and leads of CST in WHO and Autism Speaks. 

Comment 3. Were there cases where participants were both a researcher and a clinician? If so, how was it decided where they were assigned in the table? 

Thank you for raising this important point. We clarified this in the Participants section of Phase 1 as follows: 

In both phases participants were assigned to stakeholder group categories according to what they identified their main role to be. Some participants may have hold dual roles (e.g. a clinician with some involvement in research). 

Comment 4. It would be useful to know a little more about the participants and their background, specifically their experience with the CST, e.g. were most of them in the starting phase, how many had completed the adaptation process etc. How many worked in NGO’s vs public service etc. While there was an excellent range, if there was a big proportion working in NGO’s for example that would be important to know. 

We added the following information about Participants in Phase 1 of the study. 

We added that six country teams were represented in the FGDs of the twenty countries present during the event: 

All participants attending the WHO CST meeting were invited to take part in FGDs. Overall, fifteen participants consented to participate, ten in the first and five in the second group, representing six country teams of the twenty countries present. 

We added the following information about participant experiences with the adaptation work: 

In FGD1 most stakeholders had already completed the adaptation process while participants in FGD 2 were only starting it. 

We added the following information about the background of participants: 

Clinicians worked in or ran non-governmental organisations (NGOs), others worked in public hospitals. Researchers were affiliated to universities. Caregivers had overlapping experiences with working on adaptations, participating in caregiver interventions, and even running their own caregiver association. Attendance during the WHO CST meeting was not geographically representative of all continents and this is reflected in participation in the FGDs. 

We added the following information about Participants in Phase 2 of the study. We added that 10 countries were represented: 

Overall, twenty-five interviews took place using online audio or videocalls (details in Table 2): twenty-three in English and two in Spanish; representing 10 countries. Eleven participants from six CST country teams had previously participated in the FGDs. 

We added further details about the participants’ background: 

Participants represented NGOs; universities; international organisations; and advocacy organisations. 

Comment 5. The researchers positionality was thoughtfully described, I would suggest it be important to describe their own background (i.e ZS and RAH), given it’s relevance and potential for bias. E.g. are both researchers or also clinicians, do they hold more than one role and was this shared with the group? 

Thank you for this point. We added ZS’s and RAH’s role in the paragraph about positionality and it now reads as follows: 

ZS is a PhD student in Psychology, RAH is an academic Psychologist researching autism in low-resource settings, and leads research on the adaptation and evaluation of the CST programme in Ethiopia (Tekola et al., 2020). Neither ZS nor RAH is clinically qualified. 

Comment 6. Results 

The results are well described and concisely articulated. The use of quotes is appropriate and adds to the section. Perhaps I missed it, but I think it would be helpful to have a brief description/explanation that data from phase 1 and 2 were amalgamated in the results section. 

Thank you for pointing this out. To clarify, we added further points to the Analysis. We highlighted that preliminary themes were developed once we finished the FGDs. We added that once the individual interviews were completed, data were merged and analysis took place using the whole dataset. The text in Analysis now reads as follows: 

Coding started upon completion of the FGDs. The first coding of FGDs was done by ZS and in discussion between ZS, RAH and CH, allowing for investigator triangulation (47). Upon completion of the individual interviews, data from Phase 1 and Phase 2 were merged and the inductive analysis continued using the whole dataset. The first three individual interviews and the two FGDs were also coded by a second coder (BT), who has extensive experience in qualitative research and has worked previously on the adaptation of CST in Ethiopia (75). The two coders then shared their codebooks, discussed preliminary themes they developed and agreed on a codebook. ZS coded the remainder of the interviews and further refined the themes and subthemes based on discussion with RAH and CH. 

Comment 7. When sharing quotes from participants, individuals were given different descriptions – e.g. ‘clinician’ or ‘caregiver and advocate’, others had more detailed descriptions, e.g. ‘trainer of master trainers’. I would suggest, if possible, more consistency, perhaps one of clinician/researcher/caregiver and then an additional description of the participant (if possible)? 

Thank you for highlighting the need for more consistency. We adjusted the manuscript and now only use the following categories to describe participants: clinician, caregiver, researcher. 

We made changes in the title of the following participants, listing here the corrected version: 

PCP202, clinician, Americas 

PCP205, caregiver, Americas 

PCP129, caregiver, Western Pacific 

PCP18, clinician, Africa 

PCP19, researcher, Eastern Mediterranean Region 

PCP126, caregiver, Americas 

PCP126, caregiver, Americas 

Comment 8. Discussion 

The discussion was considered, included relevant literature and outlined some of the complexity of this subject area. The authors at times chose to reflect opposing views expressed by the participants which added to the richness of the section. The key sections of the results are elaborated on and examined. 

Comment 9. Although the implications section raises salient points, I wonder if there could be a little more on reflections for future research in this area. Given how challenging it is to capture the role of context in adapting and implementing caregiver interventions, providing some guidance in the area would be useful. 

We added the following points on possible areas for future research to the section about Implications: 

Further research would be helpful to understand whether the core elements of caregiver-mediated interventions for DDs are culturally universal. If they are, it would then be important to investigate how contextual factors, including cultural differences in caregiver-child engagement may impact their local implementation. When new strategies are developed to explain and implement psychological concepts across settings, rigorous and systematic evaluation is needed to understand whether they remain evidence-based. 

Comment 10. Page 25, last paragraph, line 5, perhaps this could be rewritten for clarity. 

We rewrote this sentence, which now reads as follows: 

However, informants emphasised that the programme should not rely on only one person’s motivation to keep the intervention going in the long run, as it is not sustainable. A suggested example to improve sustainability was to let a governmental body take ownership of implementing the intervention. 

Comment 11. Limitations 

I think a line on the ways in which not having stakeholders who left CST participate in the study is a limitation, would be useful. 

There was one participant who left CST and we mentioned this in the very beginning of Limitations. However, our phrasing might have been unclear. We therefore clarified this sentence and it now reads as follows: 

A further limitation is that the perspectives of stakeholders who leave the CST or stop running caregiver interventions were underrepresented: there was only one stakeholder interviewed who left working on the CST programme. 

Comment 12. Is a further limitation of the study that it only included stakeholders directly involved in CST? I would think that gathering information from those not involved in the programme (and outside the structure and framework of WHO) may gather further, and diverse information. Given this was a CST meeting, I also wonder if, as was alluded to, there may not have been some bias, that participants may have felt a need to be largely positive about the ways in which CST was managing adaptations and implementation (e.g. not emphasising those areas it was not managing so well). 

Thank you for highlighting these limitations. We now added them to the relevant Limitations section: 

A limitation of this study is that all participants were stakeholders directly involved in adapting or implementing the CST programme. Including voices of those working with different interventions, perhaps also some who may deliberately have chosen a different intervention because they are critical of the CST programme may have provided more diverse information. As the FGDs took place during a WHO CST meeting, participants may not have felt comfortable sharing negative, or differing perspectives about the CST programme. 

Comment 13. References 

I would suggest a look over the references for consistency, I noted for example #’s 8, 13, 58 and 62 

We reviewed the reference list to ensure that it is complete and correct. We made changes to the following references: 

8. De Leeuw A, Happé F, Hoekstra RA. A conceptual framework for understanding the cultural and contextual factors on autism across the globe. Autism Res. 2020;13(7):1029-1050. 

13. Dawson-Squibb J-JS. Parent education and training for autism spectrum disorder: evaluating the evidence for implementation in low-resource environments. 2018; Available from: https://open.uct.ac.za/bitstream/handle/11427/28355/thesis-hsci_Dawson_2018.pdf?isAllowed=y&sequence=1

62. López N, Gadsden VL. Health inequities, social determinants, and intersectionality. 2016;1–15. 

We found reference number 58 to be correct. We added the following references upon this revision: 

73. Smythe T, Zuurmond M, Tann CJ, Gladstone M, Kuper H. Early intervention for children with developmental disabilities in low and middle-income countries–the case for action. International Health. 2021;13(3):222–231. 

74. Shih W, Shire S, Chang YC, Kasari C. Joint engagement is a potential mechanism leading to increased initiations of joint attention and downstream effects on language: JASPER early intervention for children with ASD. Journal of child Psychology and Psychiatry. 2021;62(10):1228–1235. 

75. Tekola B, Baheretibeb Y, Roth I, Tilahun D, Fekadu A, Hanlon C, Hoekstra R. Challenges and opportunities to improve autism services in low-income countries: lessons from a situational analysis in Ethiopia. Global mental health. 2016;3:e-21. 

76. Zeidan J, Fombonne E, Scorah J, Ibrahim A, Durkin MS, Saxena S, Elsabbagh M. Global prevalence of autism: A systematic review update. Autism Research. 2022;1-13. 

77. Hoeft TJ, Fortney JC, Patel V, Unützer J. Task‐sharing approaches to improve mental health care in rural and other low‐resource settings: a systematic review. The Journal of rural health. 2018;34(1):48–62. 

78. Padmanathan P, De Silva MJ. The acceptability and feasibility of task-sharing for mental healthcare in low and middle income countries: a systematic review. Social science & medicine. 2013;(9):82–86. 

79. Punch S. Childhoods in the majority world: Miniature adults or tribal children? Sociology. 2004;37(2):277–295. 

80. Salomone E, Ferrante C, Salandin A, Ferrara F, Torchio E, Foletti G, Servili, C. Acceptability and feasibility of the World Health Organization’s caregiver skills training implemented in the Italian national health system. Autism. 2021; 13623613211035228. 

From the editor: 

Please ensure that your manuscript meets PLOS ONE's style requirements, including those for file naming. The PLOS ONE style templates can be found at  

https://journals.plos.org/plosone/s/file?id=wjVg/PLOSOne_formatting_sample_main_body.pdf and 

Thank you for highlighting this. We removed bold and italic from the title page and we now use the * sign to identify the corresponding author. We added the following information to the title page: 

^Membership of the WHO CST Team is provided in the Acknowledgments. 

We added the Acknowledgements to the end of the manuscript: 

Acknowledgements 

We would like to thank the whole WHO CST Team for their support with this study. The WHO CST Team includes (in alphabetical order): Felicity L. Brown (War Child Holland and University of Amsterdam, Netherlands), Laura Pacione (University of Toronto, Canada), Erica Salomone (University of Milano-Bicocca, Italy), Chiara Servili (WHO), Stephanie Shire (University of Oregon, USA) 

We changed Level 1 Headings’ font size to 18 pt; Level 2 Headings to 16 pt; and Level 3 Headings to 14 pt as per PLOS One requirements. 

We changed the Table headings to bold and added a point after the number of the table, as well as after the heading itself. 

We separated supporting information into 5 files, from S1-4 File to S1 Table. 

During our internal checks, the in-house editorial staff noted that you conducted the phase 1 of your research in another country. Please check the relevant national regulations and laws applying to foreign researchers and state whether you obtained the required permits and approvals. Please address this in your ethics statement in both the manuscript and submission information. In addition, please ensure that you have suitably acknowledged the contributions of any local collaborators involved in this work in your authorship list and/or Acknowledgements. Authorship criteria is based on the International Committee of Medical Journal Editors (ICMJE) Uniform Requirements for Manuscripts Submitted to Biomedical Journals - for further information please see here: https://journals.plos.org/plosone/s/authorship

Thanks for this note. We adjusted out Ethics statement accordingly: 

Our ethical approval allowed for data collection for Phase 1 of this study with participants on the premises of the WHO CST Consultation Meeting that took place in China. 

We ensured that we have suitable acknowledged the contributions of local collaborators. 

We reviewed the reference list to ensure that it is complete and correct. We did not cite papers that have been retraced. We made changes to some of the references and we added new ones: to complete list of these can be found in response to Reviewer 2’s Comment 13 on page 10 of this response letter. 

Additional changes: 

We changed the term developmental disabilities to developmental disabilities throughout the manuscript, according to the change in terminology by the World Health Organization. We highlighted changes in the manuscript in green. 

We hope these revisions are to the editor’s and reviewers’ satisfaction. 

Kind regards on behalf of all authors, 

Zsofia Szlamka, Corresponding author

---

## [Decision Letter · Decision Letter 1]

28 Jun 2022

PONE-D-21-29533R1Exploring contextual adaptations in caregiver interventions for families raising children with developmental disabilitiesPLOS ONE

Dear Dr. Szlamka,

Thank you for submitting your revised manuscript to PLOS ONE. The reviewers and I have carefully considered your manuscript. As you will see, there are only a few minor editorial changes that will need to be addressed. Therefore, I invite you to submit a revised version of the manuscript that addresses these points.

We look forward to receiving your revised manuscript.

Kind regards,

Eric J. Moody, Ph.D.

Academic Editor

PLOS ONE

Journal Requirements:

Reviewers' comments:

Reviewer's Responses to Questions

**Comments to the Author**

1. If the authors have adequately addressed your comments raised in a previous round of review and you feel that this manuscript is now acceptable for publication, you may indicate that here to bypass the “Comments to the Author” section, enter your conflict of interest statement in the “Confidential to Editor” section, and submit your "Accept" recommendation.

Reviewer #1: (No Response)

Reviewer #2: All comments have been addressed

2. Is the manuscript technically sound, and do the data support the conclusions?

Reviewer #1: Yes

Reviewer #2: Yes

3. Has the statistical analysis been performed appropriately and rigorously? 

Reviewer #1: N/A

Reviewer #2: N/A

4. Have the authors made all data underlying the findings in their manuscript fully available?

Reviewer #1: Yes

Reviewer #2: Yes

5. Is the manuscript presented in an intelligible fashion and written in standard English?

Reviewer #1: Yes

Reviewer #2: Yes

6. Review Comments to the Author

Reviewer #1: The authors have made detailed responses to the comments raised by the reviewers. I have only minor suggestions to make.

Abstract: 4th last sentence (Participants discussed ..) is unnecessary as it repeats two theme headings without adding information, therefore omit.

On page 5 the word 'context' is given in italics. I think it would be helpful to do the same on page 28 for the words 'tension' and 'respect'.

Page 6, line 7: An example of (not 'to')

Page 8: Table heading has mixed font.

Page 28, line 10: should this be 'joint' (not joined)?

Page 29, line 6: should this be 'benefits'? I think better to say 'receipt of financial benefits'.

Page 31, Limitations 2nd sentence, line 4: comma needed after 'programme'.

Reviewer #2: Thank you for comprehensively addressing the comments, I have no additional recommendations and believe it to be an important contribution that will increase our knowledge in this area were it to be published. Again, congratulations on your work.

7. PLOS authors have the option to publish the peer review history of their article (what does this mean?). If published, this will include your full peer review and any attached files.

Reviewer #1: No

Reviewer #2: **Yes: **John-Joe Dawson-Squibb

---

## [Author Response · Author response to Decision Letter 1]

4 Jul 2022

Reviewer #1: The authors have made detailed responses to the comments raised by the reviewers. I have only minor suggestions to make.

Comment 1. Abstract: 4th last sentence (Participants discussed ..) is unnecessary as it repeats two theme headings without adding information, therefore omit.

Thank you for the suggestion. We omitted this sentence. The Results section of the Abstract now reads as follows:

Four main themes were developed: 1) Setting the scene for adaptations; 2) Integrating an intervention into local public services; 3) Understanding the reality of caregivers; 4) Challenges of sustaining an intervention. Informants thought that contextual adaptations were key for the intervention to fit in locally, even more so than cultural factors. The socio-economic context of caregivers, including poverty, was highlighted as heavily affecting service access and engagement with the intervention. Competing health priorities other than DDs, financial constraints, and management of long-term collaborations were identified as barriers. 

Comment 2. On page 5 the word 'context' is given in italics. I think it would be helpful to do the same on page 28 for the words 'tension' and 'respect'.

We took your advice, and the words ‘tension’ and ‘respect’ are now written in italics on page 28. It reads as follows:

Examples include tension being a specific idiom of distress in India (56) or respect a specific Latin American value of parenting (57).

Comment 3. Page 6, line 7: An example of (not 'to')

Thank you for spotting this mistake. We corrected it and the sentence reads as follows:

An example of this cultural focus is using the term ‘ubuntu’ to ground early interventions for DDs in Africa (75).

Comment 4. Page 8: Table heading has mixed font.

Based on your comment we made sure that the table heading has a consistent font on Page 8 and it now reads as follows:

Table 1. Characteristics of participants in the focus group discussions

Comment 5. Page 28, line 10: should this be 'joint' (not joined)?

Thank you for spotting this error. We changed ‘joined’ to ‘joint’ and the sentence now reads as:

For example, a previous autism intervention study in the USA suggested up to 69% of the intervention effect on language improvement may be mediated by joint engagement (74).

Comment 6. Page 29, line 6: should this be 'benefits'? I think better to say 'receipt of financial benefits'.

Thank you for the suggestion. We changed ‘benefit’ to ‘the receipt of financial benefits’. The sentence now reads as:

Caregivers may not be informed about DDs (believing that DDs can be cured) and they may not have accurate information about their rights (when would the receipt of financial benefits stop).

Comment 7. Page 31, Limitations 2nd sentence, line 4: comma needed after 'programme'.

We added the comma to the second sentence of the Limitation section. It now reads as:

Including voices of those working with different interventions, perhaps also some who may deliberately have chosen a different intervention because they are critical of the CST programme, may have provided more diverse information.

Reviewer #2: Thank you for comprehensively addressing the comments, I have no additional recommendations and believe it to be an important contribution that will increase our knowledge in this area were it to be published. Again, congratulations on your work.

Thank you for providing your thoughts and feedback.

---

## [Decision Letter · Decision Letter 2]

13 Jul 2022

Exploring contextual adaptations in caregiver interventions for families raising children with developmental disabilities

PONE-D-21-29533R2

Dear Dr. Szlamka,

We’re pleased to inform you that your manuscript has been judged scientifically suitable for publication and will be formally accepted for publication once it meets all outstanding technical requirements.

Kind regards,

Eric J. Moody, Ph.D.

Academic Editor

PLOS ONE

Additional Editor Comments (optional):

Reviewers' comments:

Reviewer's Responses to Questions

**Comments to the Author**

1. If the authors have adequately addressed your comments raised in a previous round of review and you feel that this manuscript is now acceptable for publication, you may indicate that here to bypass the “Comments to the Author” section, enter your conflict of interest statement in the “Confidential to Editor” section, and submit your "Accept" recommendation.

Reviewer #1: All comments have been addressed

2. Is the manuscript technically sound, and do the data support the conclusions?

Reviewer #1: Yes

3. Has the statistical analysis been performed appropriately and rigorously? 

Reviewer #1: N/A

4. Have the authors made all data underlying the findings in their manuscript fully available?

Reviewer #1: Yes

5. Is the manuscript presented in an intelligible fashion and written in standard English?

Reviewer #1: Yes

6. Review Comments to the Author

Reviewer #1: (No Response)

7. PLOS authors have the option to publish the peer review history of their article (what does this mean?). If published, this will include your full peer review and any attached files.

Reviewer #1: No

---

## [Editor Report · Acceptance letter]

15 Jul 2022

PONE-D-21-29533R2 

Exploring contextual adaptations in caregiver interventions for families raising children with developmental disabilities 

Dear Dr. Szlamka:

I'm pleased to inform you that your manuscript has been deemed suitable for publication in PLOS ONE. Congratulations! Your manuscript is now with our production department. 

Kind regards, 

on behalf of

Dr. Eric J. Moody 

Academic Editor

PLOS ONE